# Impact of Germination on the Microstructural and Physicochemical Properties of Different Legume Types

**DOI:** 10.3390/plants10030592

**Published:** 2021-03-22

**Authors:** Denisa Atudorei, Silviu-Gabriel Stroe, Georgiana Gabriela Codină

**Affiliations:** Faculty of Food Engineering, Stefan cel Mare University of Suceava, 720229 Suceava, Romania; denisa.atudorei@outlook.com (D.A.); silvius@fia.usv.ro (S.-G.S.)

**Keywords:** legume, germination, lyophilization, microstructure, chemical compounds, FT-IR analysis, principal component analysis

## Abstract

The microstructural and physicochemical compositions of bean (*Phaseolus vulgaris*), lentil (*Lens culinaris Merr.*), soybean (*Glycine max* L.), chickpea (*Cicer aretinium* L.) and lupine (*Lupinus albus*) were investigated over 2 and 4 days of germination. Different changes were noticed during microscopic observations (Stereo Microscope, SEM) of the legume seeds subjected to germination, mostly related to the breakages of the seed structure. The germination caused the increase in protein content for bean, lentil, and chickpea and of ash content for lentil, soybean and chickpea. Germination increased the availability of sodium, magnesium, iron, zinc and also the acidity for all legume types. The content of fat decreased for lentil, chickpea, and lupine, whereas the content of carbohydrates and pH decreased for all legume types during the four-day germination period. Fourier transform infrared spectroscopic (FT-IR) spectra show that the compositions of germinated seeds were different from the control and varied depending on the type of legume. The multivariate analysis of the data shows close associations between chickpea, lentil, and bean and between lupine and soybean samples during the germination process. Significant negative correlations were obtained between carbohydrate contents and protein, fat and ash at the 0.01 level.

## 1. Introduction

Today, we are witnessing continuous progress at all levels. This is why it is necessary to synchronize the trends in the food industry with those of today, which should, of course, include the requirements and preferences of consumers. What is of increasing interest nowadays is the concept of innovation, which is required more and more by the food industry. In this field, the concept is related to the obtention of new food products that may satisfy and attract as many categories of consumers as possible. The possibility of using different legume types in various forms in order to create innovative foods such as pasta [1], yoghurt [2,3], bakery products [4,5], drinks [6], etc., is an increasing trend nowadays that may satisfy the consumer demand. Studying the consumer market and also the literature, it was observed that consumer interest in healthy diets has increased in recent years. This is a result of the fact that more and more people nowadays face various food deficiencies or various health conditions caused by inadequate nutrition [7,8,9,10,11]. By closely studying what people purchase from pharmacies, it was observed that more and more people buy food supplements based on vitamins and minerals [12]. However, these are produced in laboratories by chemical methods. However, nutritionists recommend that the population adopt a balanced diet that fully meets the vitamin and mineral needs of consumers of all ages. Unfortunately, even if the population understands this, it is not always possible to adopt a healthy lifestyle because some foods, even basic ones, are poor in nutrients and rich in high-calorie compounds [13,14,15,16].

Considering the fact that legumes have a balanced nutritional composition, being an ideal source of minerals, vitamins, proteins, etc. [17,18], but also the fact that they help prevent and treat various diseases [19,20], their use for consumption would be desirable. However, they are also known to contain a number of antinutritive compounds [21,22,23], which prevent the absorption of nutrients into the body. A suitable solution to decrease or even to eliminate the antinutritive compounds from legumes would be to use the germination process to balance them. Studies in the field have repeatedly highlighted, through concrete data, the advantages of the germination process over the nutritional profile of legumes [24,25]. If we also think about the fact that germination is a relatively simple process that does not require special uses and techniques and is also friendly to the environment, then the preference for this process increases. However, studies conducted so far show that the number of advantages depends on how the germination process is conducted. Therefore, in the first phase, it would be necessary to carefully study the influence of the germination process on legumes in order to establish an optimum process.

The importance of this study is supported by the idea that germination is of particular interest in the food industry. Therefore, there are a lot of studies in the literature that have highlighted the possibility of using different legumes in germinated form or flours from them, emphasizing the influence of this addition, both in terms of optimizing the nutritional profile of the food products and on their quality characteristics. Thus, the authors of various articles have outlined the fact that legumes in a germinated form can be used as an addition (as such or in powder form) in a wide range of foods, such as drinks [26], sweet products [27], various salty foods, bakery products, dairy products [28], pasta [29], crackers [30], etc. Depending on the food products, the addition of legumes in a germinated form has different effects. Therefore, studies are needed to highlight all these effects, but these studies should first start by highlighting all the transformations that take place in the grain during germination to better understand how the addition of it in a germination form would influence the food product quality.

Germination is an advantageous process that can be used successfully to improve the nutritional profile of grains subjected to this process, as evidenced by studies in the field [31,32]. This is of interest process because grains in a germinated form can be incorporated as such or in powder form in various recipes of many categories of foods in order to improve their sensory and nutritional quality, but also to reduce the number of chemical additives used. For example, they can be used successfully to replace the addition of enzymes in the bread-making process because during germination the enzymes in the grain are activated, having an essential role in producing bakery products of a high-quality. This is an example of the possibility of using these grains in a germinated form, but the examples can continue with other food products.

This study highlights the influence of the germination process on the internal profile of the different types of legumes (bean, lentil, soybean, chickpea and lupine) and on their nutritional profile (minerals, lipids, proteins, ash) before the germination process begins, at two days and four days, respectively, of the germination period. The germination process effects on different types of legume seeds were analyzed using different modern instruments such as a stereo microscope, scanning electron microscope (SEM), atomic absorption spectrometry, Fourier transform infrared spectroscopic (FT-IR) spectrometer, etc., on nongerminated and germinated seeds for 2 and 4 days. All this must be highlighted in the literature because, before the legumes in the germinated form are used as an addition in the recipes for the manufacture of various foods, it is necessary to understand the germination mechanism and the physical and physiological transformations in the grain, so that the germination form to be used with success, depending on the desired goal. The study is of particular interest due to the fact that, to our knowledge, in the literature there are not many publications related to the germination of different legumes, especially since each legume undergoes specific changes during the germination process.

## 2. Results

### 2.1. Appearance of Legume Seeds during Germination Period

Figure 1 shows the images captured with the stereo microscope device before subjecting the legume seeds (bean, lentil, soybean, chickpea and lupine) to the germination process and during the second and the fourth days of the germination period. 

### 2.2. SEM Analysis

In Figure 2a, the microscopy images (SEM) of raw bean, lentil, soybean, chickpea, and lupine seeds can be seen. These were similar to those published in the literature [33,34,35,36,37].

### 2.3. Physical-Chemical Characterization of Legume Seeds during the Germination Period

The results from Table 1 show that proteins in legume seed flours varied between 19.4 and 40.3% (dry weight), this amount being high due to the fact that the legume seeds are among the richest sources of proteins for animal and human nutrition [38]. Soybean and lupine flours are richer in protein compared to the bean, lentil and chickpea flours, which are around the 20–30% value. Similar data for the protein content of these legume types has also been reported by different researchers [39,40]. Bean, chickpea and lentil flours in germinated form presented an increase in total protein content compared to nongerminated seeds flour. As may be seen from the data obtained, germination for up to four days increased the total protein content by 15.04% in beans, by 8.76% in chickpea and by 3.14% in lentil. Similar results were also reported for germinated bean [41,42,43], chickpea [44,45,46] and lentil seeds [45]. However, for lupine flour the total protein content value decreased by 1.25%, whereas for soybean flour any significant variation (*p* < 0.05) during the four-day germination period was not noticed. 

For soybean, the lipid content increased in a significant way (*p* < 0.05), especially after 2 days of germination period; these data are in agreement with those reported by [47]. Considering all legume types analyzed, soybean presented the highest fat contents. It is a species of legume that it is classified as an oilseed due to its high fat content. It is well known that oil crop fat accumulation takes place during maturation and seed development. Regarding the variation of lipid content from the bean flour, it may be seen that its content did not significantly change (*p* < 0.05) during the germination process. 

Ash content of nongerminated seeds ranged from 2.6% in lentil flour to 4.5% in soybean flour. On germination, there was a significant increase (*p* < 0.05) in ash content for lentil, soybean and chickpea flours. However, for bean flour the ash content decreased, whereas for lupine flour the ash content did not present any significant variations (*p* < 0.05) during the germination process. 

The moisture content in nongerminated flour samples ranged from 7.6 to 10.9%. After germination and lyophilization, the moisture content of the germinated seed flour did not exceed the 10.5% value, a fact that allowed us to store them for a longer period of time in order to be used as ingredients in different food products. 

Carbohydrate contents after germination decreased for all the analyzed samples, except lupine, whose value did not vary in a significant way (*p* < 0.05). However, seed types whose protein levels decreased during the germination process, such as lupine, had slightly increased carbohydrate contents. 

The pH decreased and acidity increased in a significant way (*p* < 0.05) for all seed samples during the germination period.

Sodium, magnesium, iron and zinc contents increased for all the seed samples during the germination period, as may be seen in Table 2. Germination led to an increase in sodium and magnesium for all types of legumes during the germination period. This increase was significant (*p* < 0.05) for all seeds in the four-day germination compared to the control samples. The beneficial influence on iron and zinc contents was similar to that of sodium and magnesium for all germinated seed samples. However, this increase was not significant (*p* > 0.05) one for legume seeds in the case of zinc and for bean in the case of iron during the germination period.

### 2.4. FT-IR Analysis

Analyzing the spectra shown in Figure 3, which were obtained using Fourier transform infrared spectroscopy, it can be seen that the compositions in terms of chemical compounds for nongerminated legume seeds and seeds subjected to germination for 2 days and 4 days, respectively, were different. From the analysis of Figure 3, it can be concluded that in the studied spectral range (4000–800 cm^−1^), there were several peaks that correspond to the different molecular bonds of chemical compounds (proteins, lipids, carbohydrates) that interact with infrared radiation. In the literature, it was highlighted that the spectral range between 3000 and 2800 cm^−1^ corresponds to lipid compounds, due to vibrations produced by carbonyl groups of triglycerides (C-H) [48,49,50].

### 2.5. Relationships between Physico-Chemical Values of Legume Seeds during the Germination Period

In order to underline the correlations between the physico-chemical values of legume seeds during the germination period, PCA was used. As can be seen in Figure 4, the first two principal components (PCs) explain 66.25% (PC1—44.26% and PC2—21.99%) of the total variation. The plot of PC1 vs. PC2 loadings shows a close association between soybean and lupine samples, as well as and between chickpea, lentil and bean samples during the germination period. This may be explainable since soybean and lupine contain less carbohydrates and more proteins than the others legume samples analyzed. The carbohydrate parameter, which is predominant in the contents of chickpea, lentil and bean samples, is the closest placed to them and in opposition to the soybean and lupine samples of which protein are closely situated. Additionally, a close association was noticed between nongerminated lentil, bean and chickpea seeds, probably due to the fact that they present more similar compositions than those of nongerminated soybean and lupine seeds.

## 3. Discussion

### 3.1. Appearance of Legume Seeds during Germination Period

In the case of beans, an increase in the volume of the grain due to the fact that it absorbed water can be clearly seen. On the second day, the radicle developed up to 2.5 mm, which increased by five times at the fourth day of the germination period. In the four day of the germination period, it may also be seen that the root of the future plant developed. All these physical and physiological changes of the bean seeds during the germination period are in accordance with those described in the international literature by different researchers [51,52].

The influence of germination on the lentil development was also in agreement with different studies reported in the international literature [53]. On the second day of germination, it was clearly seen that the lentil increased in volume and began to develop the radicle, which had a size of 2.5 mm. On day 4 of germination, the radicle was high, the plumule had developed, and the dimensions of the two components of the germ were 6 mm for the radicle and 5 mm for the plumule. 

The development of lupine seeds during the germination process has also been previously reported by other authors [54]. As can be seen, on the second day of germination, in the case of lupine, the radicle was 3 mm in size. Starting on the fourth day, the germs began to synthesize chlorophyll, the radicle size being 11 mm. 

In the case of chickpea germination, on the second day of germination period seed radicle development was clearly seen, of which the size was of 3 mm. In the fourth day of the germination period, the radicle size increased five times more, its length being of 16 mm. Additionally, on the fourth day of germination period how the leaves of the future plant developed was clearly seen. 

For soybean, it can be said that radicle development began on the second day of germination and had the size of 3 mm on this day. The image from the second day of soybean germination clearly shows the degradation of the outer shell of the grain in order to allow the development of the radicle. In the image showing the fourth day of germination, it can be seen how the leaf developed inside the bean. The size of the radicle reached 24 mm in length on this day of germination. Similar changes during soybean germination have also been reported by different researchers [55,56].

### 3.2. SEM Analysis

Analyzing the microscopy images (SEM) of raw bean, lentil, soybean, chickpea and lupine seeds, middle lamellas separating seed cells and cell walls were clearly visible in all images of nongerminated seeds. In the case of bean, chickpea and lentil, cotyledon cells composed of protein materials which cover elongated starch granules were seen. In the case of soybean and lupine seeds, which contain less starch, more dense compactness of the protein agglomerates was visible.

In germinated seeds, significant structural changes may be seen in cotyledon cells for all analyzed samples. Middle lamellas separating seeds cells and cell walls were visible. In the case of chickpea and bean, a high number of depicted starch granules, spherical ones with smooth surfaces covered in some parts by amorphous protein structures, could clearly be seen. As the germination period went on, cell structures from the germinated seeds continued to change. It may be seen that starch granules were still separated by proteins and presented smooth surfaces but with higher disruption of the cellular order, probably due to the increased activity of the hydrolytic enzymes. More tiny filaments between starch granules and their neighboring proteins appeared. Furthermore, some precipitate was seen on starch granules, probably due to the increase in the proteolytic activity with the increase in the germination period [36]. With the germination process, the changes were more pronounced in the cotyledon cells. Some breakages of the surface were clearly distinguishable. Lupine seeds, as well as soybean and also lentil seeds, presented a different microstructure during germination than those of bean and chickpea. These were probably due to their higher content of protein and lower amount in starch in the case of soybean and lupine seeds. Spherical to oval protein bodies were visible in the continuous cell matrix which appeared to be denser materials. After germination, the cells were more loosely packed with large intercellular spaces between them. These intercellular spaces varied from a few in nongerminated seeds to many in seeds where the germination period was longer. A shorter germination period led to a higher compactness of the protein structure, whereas a longer germination period led to a more open structure of protein agglomerates. The protein structure appeared to be granular with multiple cracking caused by seed germination, which was only slightly visible in the case of nongerminated seeds. The middle lamella, which is a pectin layer, absorbs water during germination and expands, being more visible to the germination seeds [37]. It can be clearly seen that, after germination, ruptures appear around the cells, which makes visible the middle lamella and the cell walls. Secondly, the cracks presented on the seed coat surface may be related to the water uptake during the germination process. Crack distribution may also coincide with the sites of initial water uptake [34]. After lyophilization, the water from the germination process was removed, leaving many cracks in the seeds’ structures, which were clearly visible in all the germinated seed samples.

### 3.3. Physical-Chemical Characterization of Legume Seeds during the Germination Period

The increase in the amount of protein after germination may be due to the fact that, during this process, hydrolysis of some proteins, which are inaccessible within the nongerminated seeds structure, may take place [56]. In legume seeds, the proteins are mainly localized to the seed cotyledon tissues but also to the embryonic axis and testas, which contribute less to the total protein content, mainly due to the fact that these components represent small amounts of the seed mass [57]. During seed development, stored proteins are hydrolyzed to provide carbon skeletons, nitrogen and energy required for growth and protein synthesis [58]. The proteins stored in the vegetative cells are used for seed formation, whereas the embryonic proteins stored are used for seed germination [56]. Protein release from the cell structure during the germination process varies between seed type. However, in the case of lupine and soy, the influence on the amount of protein can be attributed to a stronger hydrolysis effect on storage proteins which causes their degradation and loss for seed development [40]. It must be mentioned that the small protein variation in the case of soy may be better highlighted by the water-soluble protein through FTIR analysis [59]. Studies indicate that during the germination process the solubility of proteins increases for enzymatic activity and the development of the new tissues (the components of germs). Therefore, the amount of water-soluble proteins increases due to the germination process [56]. In this regard, FTIR analysis (presented in Section 2.4) highlighted in a better way how the amount of water-soluble protein varies depending on the increase in germination time.

In general, the germination of the legume seeds decreased the lipid content during the germination period. These data are in agreement with other studies which have also reported a decreased lipid content of some legume seeds during the germination period [42,43]. This decrease may be due to the fact that the lipid was used as an important source of energy during the sprouting process necessary for the development of seed embryos during the germination process [42]. The reduction in the lipid content of germinated flours may be an advantage for their storage due to the fact that may improve its stability [43]. In germinating soybean seeds, also characterized by high protein contents, the main metabolites in the first stage of germination may be amino acids which are important respiratory substrates and may be used in many anabolic processes, and may cause a lower breakdown of storage lipid content [39]. Additionally, even if an increase in free fatty acids and a decrease in triacylglycerol take place during the germination process, some studies [60] have reported that some complex lipids, such as phospholipids, can be synthesized during germination. These facts may lead to a constant or a slight increase in the total lipid content, especially in the first stage of the germination period. In the case of bean, a nonsignificant change of lipid content during the germination process has also been reported by Ferreira (2019) [46] for chickpea germination over a 48 h period. According to this study, this behavior may be due to the short germination period reported for this seed type, which presented the highest size from all the analyzed legume types. Lipids are polymeric structures used for storage, mainly in the form of triacylglycerol. In order to be used for energy generation, the lipases must act on triglycerides to release free fatty acids. In the case of beans, the use of lipids may not have been so high during the 4 days of germination. 

The significant increase (*p* < 0.05) in ash content for lentil, soybean and chickpea flour could be due to the increase in phytase activity during germination, which hydrolyzed the bond between the proteins, enzymes and minerals, to release the minerals [43,61]. The decrease in the amount of ash in the case of beans may be due to a shorter germination period reported to the seed type. A reduction in the ash content during the germination period has also been reported by Ferreira (2019) [46] for chickpea germination during the 48 h period. It seems that the ash variation during the germination process depends on the grain type and the germination period [42,62]. Different studies reported that the ash value presented the highest value after 72 h of germination period, the time used by us for the legume germination [63,64]. Additionally, some studies reported that the increase in ash content occurs due to the reduction in total soluble solids [65]. This is probably due to hydrolytic enzymes that have not favored the accumulation of total soluble solids, which led to their increased levels [66].

The reduction in carbohydrate contents after germination may be attributed to amylases activity which increases during germination, leading to a starch degradation process [67]. Additionally, this decrease may be due to the carbohydrates use as a substrate during the germination period leading to an increase in the protein levels [46]. However, the amount of carbohydrate variation during the germination process depends on the legume type probably due to a stronger or weaker hydrolyses of them. During the germination period, the carbohydrates were solubilized in order to support the development of the components of the germs [56]. 

The pH decrease and acidity increase could be due to the activity of the lipase, which acts on triacylglycerols converting them into free fatty acids necessary for energy generation [46].

The increase in the availability of the analyzed minerals was a general effect of the germination process, which is related to the phytase action, phytate content, extent of binding of minerals within the matrix, or interaction of these factors. Legumes are rich in α-amylase inhibitors, polyphenolic compounds, protease inhibitors, tannins, lectins and phytic acid that cause poor absorption and digestibility of minerals and nutrients. More, legumes contain phytase enzymes that are activated by germination to destroy phytate [68]. Phytate is found in legumes in the form of phytic acid and is considered to be a chelating agent because it forms phytates with minerals. During germination, there is a decrease in the amount of phytate due to its hydrolysis by the enzyme phytase [69]. Phytase is a phytate-specific phosphatase. This hydrolyzes phytate to inositol and free orthophosphate and releases minerals [70]. This may be a reason why the amount of minerals increases after germination. Studies have also shown that with the increase in the amount of minerals after germination, there is an improvement in their digestive availability. Therefore, in the longer germination period, the legumes’ mineral contents increase. These data agree with those obtained by different researchers or various types of vegetables during the germination process [71,72]. The literature does not indicate particular mechanisms for each mineral substance that may elucidate the increase in the amount of this following the germination process. Each increase is attributed to the three processes—namely, the decrease in the amount of phytic acid, the activation of phytase and the release of minerals found in legumes in bound form (in the constitution of different complex substances) [65,68,70,73,74]. These transformations occur due to the fact that germination is a catabolic process by which, in the first stage, with the absorption of water, the reserve substances from seeds are hydrolyzed in order to provide nutrients important for the development of germs. A decrease in phytic acid content lad to an increase in some mineral contents from the seed subjected to germination [75]. However, all these variations are different for each grain.

### 3.4. FT-IR Analysis

By analyzing the spectra obtained for bean in both raw and germinated forms, it can be seen that the highest absorbance at the wavelength range specified above was recorded in the case of nongerminated beans and the lowest absorbance value was recorded in the case of beans that germinated for two days. The peaks recorded in this spectral range are also due to the vibrations of the -C-H(CH_2_) and -C-H(CH_3_) groups in the composition of fatty acids, as explained in the literature [50]. Additionally, the literature highlights the fact that the peaks recorded around the wavelength 1750 cm^−1^ are determined by the C=O bonds of the ester or carboxylic acid groups, in the FT-IR spectra this peak is in the range of 1800–1700 cm^−1^. The spectral range between 1700 and 1500 cm^−1^ corresponds to the protein content of the samples. Analyzing the spectra obtained in the case of beans, it can be concluded that the absorbance recorded a higher value in the case of beans subjected to germination for 4 days and the lowest value for nongerminated beans. The two peaks that are very clearly visible are suitable for amide I (C=O, C-N), with a wavelength of about 1650 cm^−1^, and for amide II (N-H, C-N), with a wavelength of about 1550 cm^−1^, as shown by the interpretation of the bean spectra. Additionally, the second peak recorded is due to the peptide (CO-NH) bond [76,77]. Thus, it can be concluded that the peaks in the range of 1700–1500 cm^−1^ are due to the vibrations produced by the peptide bonds and thus they can be an indicator for the protein content of the samples. To characterize the carbohydrates in the samples using FT-IR determination, different researchers have showed that the wavelengths in the range 1200–900 cm^−1^ correspond to carbohydrates [78]. By analyzing the obtained spectra, it can be concluded that the absorbance recorded a higher value in the case of nongerminated beans and the lowest value for two days of bean germination. The highest intensity of the peak was recorded for the wavelength of about 1050 cm^−1^, which corresponds to the OH group of carbohydrates. It was also observed that the peaks differ more in intensity values, not in the shapes of the peaks.

Analyzing the spectra obtained for nongerminated lentil and for the lentil germinated (two and four days), it can be concluded that the wavelength between 1700 and 15,000 cm^−1^ corresponds to the amount of protein. The maximum height of the peak was obtained for the wavelength of 1700 cm^−1^. The recording of the peak at this wavelength is due to the vibration made by the OH and C=O groups in the amide I region [79,80], which provides indications of the protein content of the sample. The second peak in this spectral region is due to the vibration caused by the N-H and C-N groups associated with amide II [81]. The wide peak corresponding to the maximum wavelength of 1050 cm^−1^ corresponds to the vibration of the C-O groups from the carbohydrate content [80,82].

From the analysis of the obtained spectra for the nongerminated and germinated lupine grains (at two days and at four days of germination, respectively), it can be concluded that the spectra for nongerminated lupine and those for 2 days and 4 days of germination, respectively, are similar—only the value of absorbance differs. For example, in the spectral range of 1200–800 cm^−1^, there were peaks with high absorbance values in the case of germinated chickpeas for 4 days. In this spectral range, peaks provide indications of carbohydrate concentration [83]. In this case, the values recorded correspond to those indicated by us through physicochemical data.

The spectral range between wavelengths 1700 and 1300 cm^−1^ provides indications of the amount of protein. The maximum value of absorbance was recorded in the case of nongerminated soybeans, with a wavelength value of approximately 1650 cm^−1^. The lowest peak height in this spectral range was recorded for soybeans germinated for 4 days. Regarding carbohydrate compounds that correspond to the wavelength between 1200 and 1900 cm^−1^, and according to the spectra obtained, it can be concluded that the peak had a higher value in the case of nongerminated soybeans and a lower value with the increase in the germination period. Thus, it can be concluded that the amount of carbohydrates decreased with an increasing germination period. The recording of peaks in this spectral range was achieved due to the vibrations of the C-OH and C-O-C groups [84]. The interpretation of the data obtained by using FT-IR analysis corresponds to the physicochemical data which also indicated that in the case of germinated soybeans the amount of protein and carbohydrates was lower when the germination period increased.

In the case of chickpeas, from the spectra obtained from the FT-IR analysis, it may be clearly seen that in the wavelength range between 3000–2850 cm^−1^, corresponding to lipid compounds [85], the highest value of the absorbance was recorded for nongerminated chickpea, and the value decreased with a longer germination period. Previous studies indicate similar data, namely that the peaks from the spectral range 3050–2800 cm^−1^ are due to the CH_3_ and CH_2_ groups which are in the composition of hydrocarbons from the lipid constitution. The wavelength around 3000 cm^−1^ corresponds to the vibration of the groups =C-H from the composition of fatty acids [86]. Comparing the data obtained from the FT-IR analysis and the physicochemical ones, it can be concluded that there is a correlation between the data obtained, in the sense that by physicochemical determinations it was also shown that the amount of lipids decreased with increasing germination time in the case of chickpea seeds.

### 3.5. Relationships between Physico-Chemical Values of Legume Seeds during the Germination Period

The first main component (PC1) opposes the pH and acidity values, obtaining a significant reverse correlation (r = −0.776) for a level of 0.01. This fact is explainable since it is well known that the higher the acidity, the lower the pH. Additionally, the carbohydrate contents, which are in predominant in all legume types analyzed, are in opposition with all chemical data of protein, fat and ash, which present high significant negative correlations of r = −0.930, r = −0.922 and r = −0.809 for a level of 0.01. It is known that nitrogenous substances accumulate in the grains before the starch. The more starch that accumulates in the grain during the maturation process, the lower its contents of protein and other substances are. Of the carbohydrates content, starch is predominant, and therefore the negative correlation between carbohydrates and protein, fat and ash is somehow predictable. Closeness of moisture and iron (Fe) to the center of PC shows that these variables are not useful when describing the differences between legumes analyzed. As for the second component (PC2), the lupine and soybean legume types were placed in the left part of the graph, while chickpea, bean and lentil were placed in the right part of the graph, with very good correlations shown between these samples. From the mineral association point of view with the analyzed legumes, it seems that magnesium (Mg) and zinc (Zn) minerals are more correlated with chickpea samples, which, according to the data obtained, are in the highest amount, whereas sodium (Na) is the most associated with lupine samples.

## 4. Materials and Methods

### 4.1. Materials

Bean (*Phaseolus vulgaris*), lentil (*Lens culinaris Merr.*), soybean (*Glycine max* L.), chickpea (*Cicer aretinium* L.) and lupine (*Lupinus albus*) from the 2019 crop year were cultivated in Romania and were not genetically modified according to the manufacturer declaration.

### 4.2. Germination and Lyophilization of Legume Seeds

Before the germination, the legumes were soaked for 6 h (small grains: lentils, soybeans) or for 12 h (larger grains: beans, lupine, chickpeas), in pure water, with a temperature of 20 °C in order to activate the germination process. The germination of the grains was carried out using filter paper. The germination temperature was 25 °C. The humidity of 80% was kept constant throughout the entire germination period. The germination process was carried out exclusively in dark conditions. The maximum germination time was 4 days. At the end of the germination period, this process was stopped. For this, the freeze-drying process was used because, compared to other moisture removal alternatives, this process has the least influence on the nutritional profile of germinated seed samples [87,88]. For this purpose, an Alpha 1–4 LSC plus lyophilizer was used for lyophilization. Lyophilization was performed at −50 °C, for 72 h, at a pressure of 4.2 Pa.

### 4.3. Appearance of Legume Seeds Analysis

In order to highlight the physical and physiological changes that occur in the legume profiles during the germination period, the Motic SMZ-140 stereo microscope device was used, which allowed the capturing of images with legumes in the dorsal, frontal, and ventral positions and in the section of them. For this, the stereo microscope was equipped with a Moticam 10× camera. The use of the stereo microscope made it possible to highlight the changes in legume profiles: increase in volume due to water absorption, degradation of the outer layer, the appearance of the radicle, root and plumule depending on legume type. The determinations were made taking into account the procedure described in the literature [89,90]. All these transformations were correlated with the determination of the size of the constituent parts of the germs (radicle and plumule), using a Modelcraft Vernier Caliper of 125 mm. Additionally, the caliper was used taking into account the procedure described in the literature [91].

### 4.4. SEM Analysis

The study of the microstructures of the vegetable samples was performed using a scanning electron microscope (SEM) Vega II LMU-Tescan (Brno-Kohoutovice, Czech Republic) equipped with a SE detector, which works only in a high vacuum environment, and a BSE detector, a scintillation detector that works in high and low vacuum environments. The Vega II LMU-Tescan microscope, used in this research, allows the study of samples in a pressure range between 10–2 Pa and 2 kPa. The vegetable samples were sectioned longitudinally in order to obtain a flat surface, were placed on an aluminum sample holder using a double-sided adhesive tape and were studied at an acceleration potential of 30 kV.

### 4.5. Physico-Chemical Composition Analysis

Protein, fat, ash and moisture contents were determined using AACC International Approved Methods 46-12.01, 30-25.01, 08-01.01 and 44-15.02, respectively. The pH was measured with a HQ30d portable pH Meter (HACK, Germany) on a slurry prepared with 10 g legume flours in 40 mL of boiled, deionized water according to official AOAC procedures (AOAC, 02-52.01). The total acidity value was determined according to Romanian standard method SR 90:2007. Carbohydrate contents of the legume samples were calculated according to the following formulas [92]: carbohydrates, % = 100 − (protein, % + fat, % + ash, % + moisture, % content of legumes).

The Na, Mg, Fe and Zn contents of the vegetable samples were analyzed by flame atomic absorption spectrometry (FAAS) (AA-6300 Shimadzu, Kyoto, Japan) equipped with air-acetylene flame. Hollow cathode lamps of Na, Mg, Fe and Zn were used. In total, 10 g with an accuracy of 10 mg from each sample was used for calcination. The calcination temperature was increased with a maximum speed of 50 °C/h up to 450 °C. The calcination time was 8 h. Ash digestion was performed using 10 mL 0.1 mol/L nitric acid (HNO_3_) (Sigma-Aldrich/Merck, Darmstadt, Germany) on a hot plate. After digestion of the ash samples, up to 50 mL was filled with bidistilled and deionized water. Standard solutions of Na, Mg, Fe and Zn (Sigma-Aldrich/Merck, Darmstadt, Germany) were used and diluted as necessary to obtain working standards. In order to eliminate the risk of contamination, all glassware was washed after each use with HNO_3_ solution and rinsed with bidistilled and deionized water. The instrumental conditions for determining the mineral content of vegetable samples by FAAS method are shown in Table 3. 

### 4.6. FT-IR Analysis

In order to highlight the changes in the composition of grains subjected to germination process, FT-IR analysis was also used. This allows the realization of a correlation between the spectra and the physicochemical data obtained previously. To obtain the spectra, a FT-IR spectrometer (Thermo Scientific, Karlsruhe, Dieselstraße, Germany) was used, equipped with the ATR IX option, which allowed the obtention of accurate data, using a detector at 4 cm^−1^. Therefore, an attenuated total reflectance-Fourier transform infrared spectroscopy (ATR-FTIR) was used to obtain the spectra. This technique is based on tracking the interaction between infrared radiation and the material under analysis. Fourier transform infrared spectroscopy (FT-IR) contains a source of light with infrared radiation which passes through the sample and this absorbs luminous energy. At the same time, occur vibrational movements due to the chemical bonds inside the molecules. These vibrational movements provide information about the chemical structure of the sample and these are provided as an FT-IR spectrum. FT-IR spectra were recorded in the spectral range situated between 800 and 4000 cm^−1^. This spectral range was suitable for characterizing the chemical compounds present in the seed samples: proteins, carbohydrates, lipids, etc.

### 4.7. Statistical Analysis

All of the data were made in duplicate and are expressed as the means of the measurements ± standard deviation. The one-way analysis of variance (ANOVA) with Tukey’s test were made to test the differences between means at a 5% significance level by using the XLSTAT statistical package (2021 version, Addinsoft, Inc., Brooklyn, NY, USA).

## 5. Conclusions

The germination period significantly influenced the growth of the seeds, leading gradually to development of the radicle and first leaves. The seed development was captured by a stereo microscope device which showed that a four-day germination may be optimum for legumes used in food consumption. The microstructures of the seeds during germination changed, ascribed in the case of bean, chickpea and lentil mainly to the starch, and in the case of lupine and soybean to the protein. Germination of seeds resulted in increased protein and ash contents for lentil and chickpea, whereas for the rest of the legume seeds their contents showed different variations. Additionally, during the germination period the fat content varied for bean and soybean, whereas it decreased for lentil, chickpea, and lupine. Mineral contents (sodium, magnesium, zinc, iron) of the germinated legume seeds increased during germination in all legume seeds, showing the beneficial influences of germination on the nutritional profile of legumes. An increase in acidity values and a decrease in pH and carbohydrate contents have also been recorded for all legume types during the germination period. Fourier transform infrared spectroscopy (FT-IR) highlighted variation in chemical compounds of legume seeds during the germination period. According to the wavelength and peak height of the FT-IR spectra, we clearly showed the fact that different compounds such as protein, carbohydrates and lipids varied depending on germination period and each legume type. Principal component analysis was performed on the combined physico-chemical and minerals data, clustering the legume types obtained during the germination process in the PC space. PCA highlighted an association between lentil, bean and chickpea samples which were placed in the right part of the graph and between soybean and lupine samples which were placed in the left part of the graph, indicating similar compositions of these samples. From the physical-chemical data point of view, significant negative correlations were obtained between carbohydrate contents and protein, fat and ash variables at a level of 0.01.

## Figures and Tables

**Figure 1 plants-10-00592-f001:**
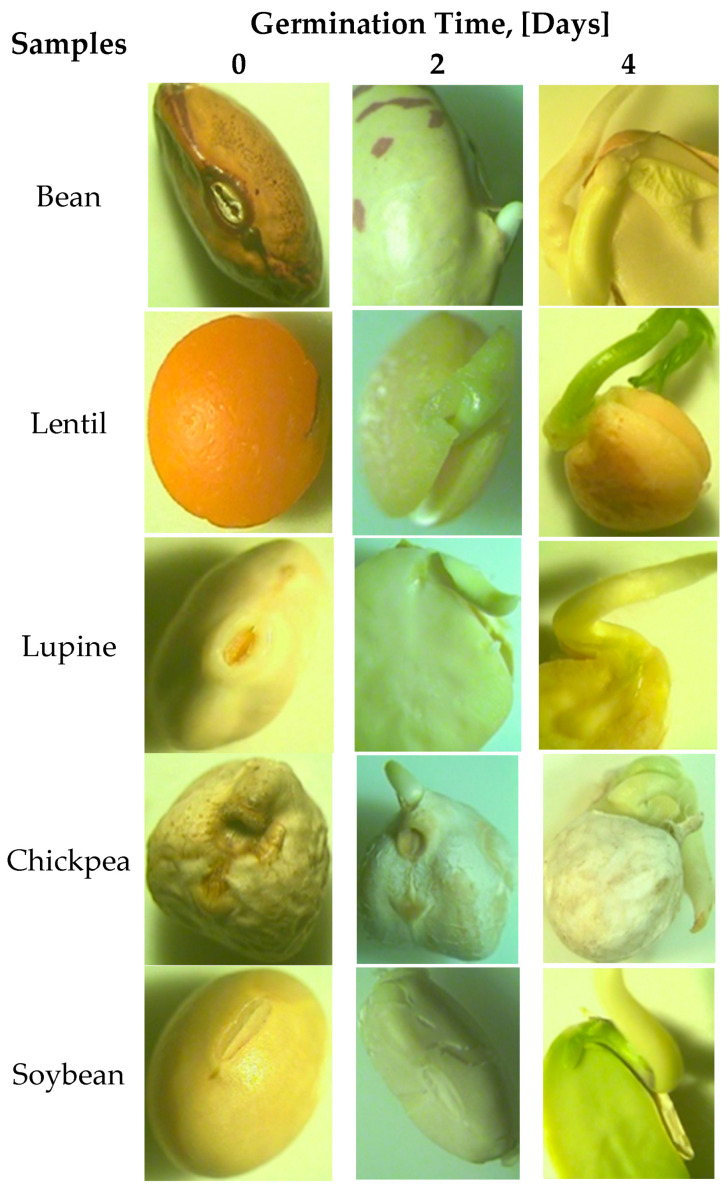
Stereomicroscope images of legumes during germination.

**Figure 2 plants-10-00592-f002:**
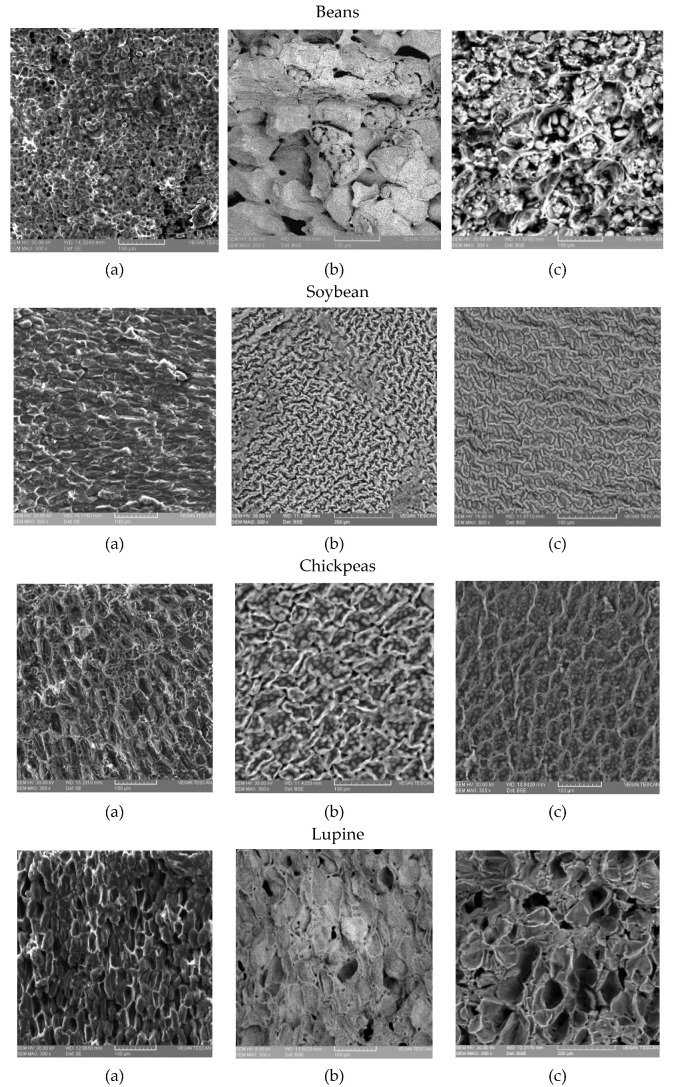
Scanning electron microscope (SEM) images showing the microstructures of legumes during germination: (**a**) nongerminated; (**b**) germinated 2 days; (**c**) germinated 4 days.

**Figure 3 plants-10-00592-f003:**
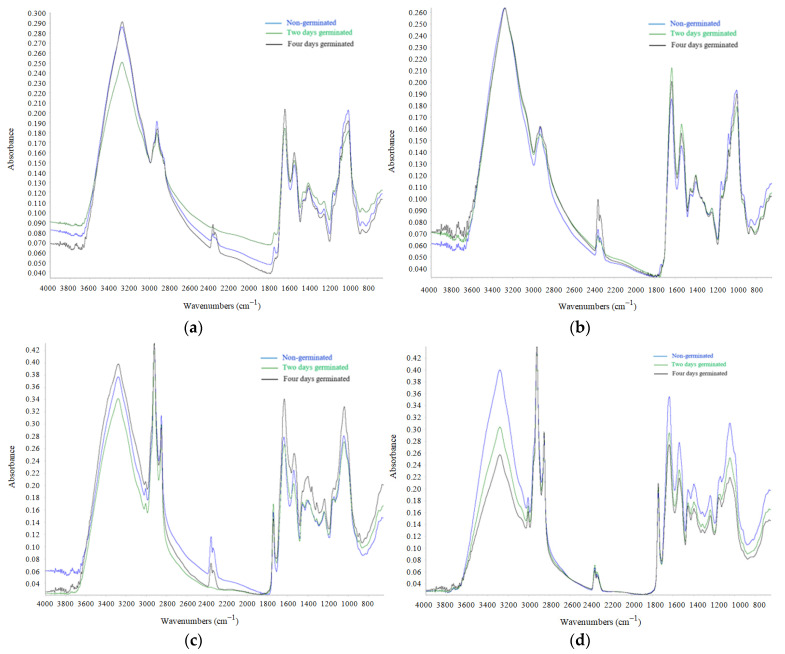
Fourier transform infrared spectroscopic (FT-IR) spectra of legumes during germination period: (**a**) bean; (**b**) lentil; (**c**) lupine; (**d**) soybean; (**e**) chickpea.

**Figure 4 plants-10-00592-f004:**
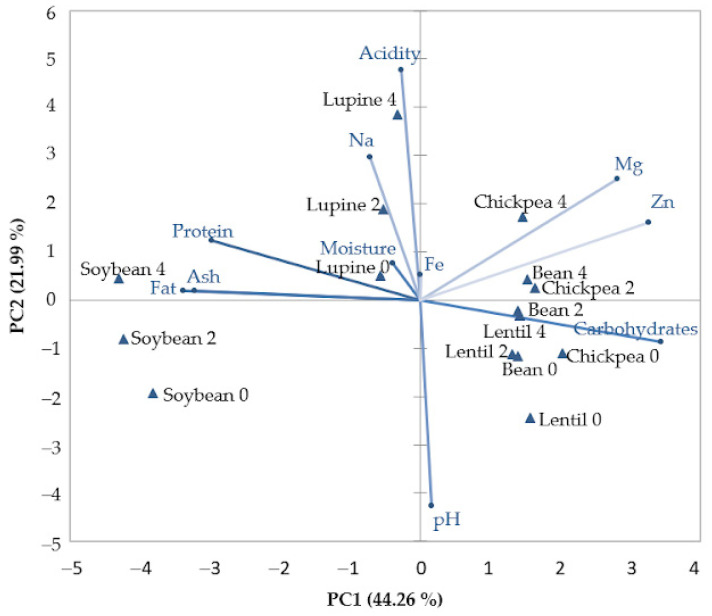
Principal component analysis for physico-chemical values of legume seeds during the germination period.

**Table 1 plants-10-00592-t001:** Physical-chemical properties of legume seeds during germination period.

Legume Type	Germination Period, Days	Protein (%)	Fat (%)	Ash (%)	Moisture (%)	Carbohydrates (%)	pH	Acidity (°)
Bean	0	22.6 ± 0.28 ^cC^	1.6 ± 0.07 ^aD^	3.6 ± 0.07 ^aB^	10.9 ± 0.07 ^aA^	61.3 ± 0.14 ^aB^	6.55 ± 0.007 ^aB^	7.15 ± 0.007 ^cB^
	2	24.1 ± 0.14 ^bO^	1.2 ± 0.14 ^aO^	3.4 ± 0.07 ^abM^	10.5 ± 0.14 ^abL^	60.6 ± 0.14 ^bL^	6.50 ± 0.021 ^aL^	8.12 ± 0.021 ^bP^
	4	26.0 ± 0.28 ^aY^	1.4 ± 0.07 ^aY^	3.0 ± 0.14 ^bV^	10.1 ± 0.14 ^bUV^	59.7 ± 0.14 ^cV^	6.41 ± 0.021 ^bV^	15.07 ± 0.056 ^aY^
Lentil	0	28.6 ± 0.28 ^bB^	1.2 ± 0.07 ^aD^	2.6 ± 0.07 ^bD^	7.6 ± 0.07 ^aC^	60.0 ± 0.07 ^aC^	6.63 ± 0.014 ^aA^	5.16 ± 0.035 ^cD^
	2	30.0 ± 0.14 ^aN^	1.1 ± 0.00 ^aP^	2.8 ± 0.07 ^bN^	8.9 ± 0.14 ^aN^	58.9 ± 0.07 ^bM^	6.51 ± 0.021 ^bL^	10.66 ± 0.021 ^bM^
	4	29.5 ± 0.14 ^abX^	1.0 ± 0.07 ^aY^	3.1 ± 0.14 ^aV^	8.8 ± 0.28 ^aX^	59.0 ± 0.14 ^bX^	6.48 ± 0.007 ^bU^	16.28 ± 0.021 ^aX^
Soybean	0	40.3 ± 0.07 ^aA^	16.6 ± 0.14 ^bA^	4.5 ± 0.07 ^aA^	9.8 ± 0.28 ^bC^	28.8 ± 0.14 ^aE^	6.62 ± 0.028 ^aA^	5.68 ± 0. 007 ^cC^
	2	40.3 ± 0.07 ^aL^	17.6 ± 0.07 ^aL^	4.7 ± 0.07 ^aL^	10.4 ± 0.07 ^aL^	27.0 ± 0.07 ^bO^	6.50 ± 0.007 ^bL^	9.83 ± 0.042 ^bO^
	4	40.2 ± 0.07 ^aU^	17.9 ± 0.07 ^aU^	5.1 ± 0.28 ^aU^	10.5 ± 0.14 ^aU^	26.3 ± 0.14 ^cZ^	6.32 ± 0.021 ^cX^	16.29 ± 0.028 ^aX^
Chickpea	0	19.4 ± 0.07 ^bD^	5.9 ± 0.14 ^aC^	3.1 ± 0.07 ^bC^	10.3 ± 0.28 ^aB^	61.3 ± 0.14 ^aA^	6.42 ± 0.007 ^aC^	4.2 ± 0.141 ^cE^
	2	20.7 ± 0.28 ^aP^	5.4 ± 0.00 ^bN^	3.5 ± 0.07 ^aM^	9.7 ± 0.07 ^abM^	60.7 ± 0.07 ^bL^	6.22 ± 0.028 ^bM^	10.24 ± 0.021 ^bN^
	4	21.1 ± 0.00 ^aZ^	5.2 ± 0.07 ^bX^	3.6 ± 0.07 ^aV^	9.4 ± 0.07 ^bVX^	60.7 ± 0.07 ^bU^	6.11 ± 0.014 ^cY^	20.45 ± 0.629 ^aV^
Lupine	0	39.9 ± 0.14 ^aA^	9.3 ± 0.21 ^aB^	3.3 ± 0.07 ^aB^	7.7 ± 0.14 ^bD^	39.8 ± 0.14 ^bD^	5.62 ± 0.007 ^aD^	14.53 ± 0.014 ^cA^
	2	39.3 ± 0.07 ^bM^	7.5 ± 0.00 ^bM^	3.3 ± 0.07 ^aM^	10.1 ± 0.07 ^aL^	39.8 ± 0.07 ^bN^	5.55 ± 0.007 ^bN^	20.87 ± 0.035 ^bL^
	4	39.4 ± 0.07 ^bV^	6.9 ± 0.07 ^cV^	3.4 ± 0.07 ^aV^	10.3 ± 0.14 ^aU^	40.0 ± 0.07 ^aY^	5.53 ± 0.007 ^bZ^	31.9 ± 0. 007 ^aU^

Means followed by the different letter (^a,b,c^) within the same column, for each legume type, are significantly different (*p* < 0.05). Means followed by different letter within the same column, for each germination period (^A,B,C,D,E^ for 0 day; ^L,M,N,O,P^ for 2 days; ^U,V,X,Y,Z,^ for 4 days), are significantly different (*p* < 0.05).

**Table 2 plants-10-00592-t002:** Minerals of legumes during germination period.

Legume Type	Germination Period, (Days)	Na (mg 100 g^−1^)	Mg (mg 100 g^−1^)	Fe (mg 100 g^−1^)	Zn (mg 100 g^−1^)
Bean	0	38.15 ± 2.61 ^bB^	141.65 ± 0.17 ^cB^	7.57 ± 0.75 ^aA^	3.22 ± 0.20 ^aA^
	2	62.62 ± 2.18 ^aL^	148.50 ± 0.11 ^bM^	7.83 ± 0.75 ^aL^	3.23 ± 0.20 ^aL^
	4	64.42 ± 0.71 ^aV^	152.15 ± 0.01 ^aX^	7.89 ± 0.80 ^aU^	3.25 ± 0.20 ^aU^
Lentil	0	26.08 ± 0.61 ^cD^	122.35 ± 0.06 ^cD^	2.62 ± 0.18 ^aC^	3.07 ± 0.20 ^aA^
	2	48.89 ± 0.94 ^bM^	126.82 ± 0.09 ^bO^	2.66 ± 0.18 ^aN^	3.10 ± 0.20 ^aL^
	4	53.52 ± 1.57 ^aX^	146.63 ± 0.12 ^aY^	2.75 ± 0.20 ^aX^	3.12 ± 0.22 ^aU^
Soybean	0	47.50 ± 1.06 ^cA^	90.30 ± 0.10 ^cE^	5.31 ± 0.14 ^aB^	1.91 ± 0.15 ^aB^
	2	60.38 ± 1.59 ^bL^	92.41 ± 0.27 ^bP^	5.32 ± 0.20 ^aM^	1.93 ± 0.10 ^aM^
	4	70.19 ± 2.03 ^aUV^	106.88 ± 0.08 ^aZ^	5.38 ± 0.21 ^aV^	1.94 ± 0.15 ^aV^
Chickpea	0	40.11 ± 1.03 ^cB^	168.31 ± 0.05 ^cA^	5.27 ± 0.30 ^aB^	3.25 ± 0.20 ^aA^
	2	58.66 ± 1.31 ^bL^	172.56 ± 0.16 ^bL^	5.29 ± 0.29 ^aM^	3.26 ± 0.18 ^aL^
	4	74.81 ± 3.50 ^aU^	174.06 ± 0.03 ^aU^	5.36 ± 0.30 ^aV^	3.28 ± 0.20 ^aU^
Lupine	0	31.94 ± 0.66 ^cC^	128.98 ± 0.02 ^cC^	4.58 ± 0.26 ^aB^	3.22 ± 0.20 ^aA^
	2	45.20 ± 1.25 ^bM^	137.75 ± 0.11 ^bN^	4.61 ± 0.20 ^aM^	3.23 ± 0.20 ^aL^
	4	69.43 ± 2.08 ^aUV^	166.63 ± 0.12 ^aV^	4.73 ± 0.35 ^aV^	3.25 ± 0.20 ^aU^

Means followed by different letter (^a,b,c^) within the same column, for each legume type, are significantly different (*p* < 0.05). Means followed by different letter within the same column, for each germination period (^A,B,C,D,E^ for 0 day; ^L,M,N,O,P^ for 2 days; ^U,V,X,Y,Z,^ for 4 days), are significantly different (*p* < 0.05).

**Table 3 plants-10-00592-t003:** Instrumental conditions for mineral analysis to flame atomic absorption spectrometry (FAAS).

Element	Wavelength (nm)	Slit Width (nm)	Fuel Gas Flow Rate (L/min)	Support Gas Flow Rate (L/min)	Flame Type	Pre-Spray Time (s)	Integration Time (s)	Response Time (s)
Na	589.0	0.2	1.8	15.0	Air-C_2_H_2_	10	5	1
Mg	285.2	0.7	1.8	15.0	Air-C_2_H_2_	10	5	1
Fe	248.3	0.2	2.2	15.0	Air-C_2_H_2_	10	5	1
Zn	213.9	0.7	2.0	15.0	Air-C_2_H_2_	10	5	1

The number of injections of the solutions in the flame was: standard solution—5; blank solution—3; sample solution—3. In order to eliminate analytical errors, each sample was analyzed three times.

## Data Availability

Not applicable.

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
