# Peer review of "Impact of Germination on the Microstructural and Physicochemical Properties of Different Legume Types"

_plants, 2021, doi:10.3390/plants10030592_

Round 1

Reviewer 1 Report

Major Comments

  1. Among parameters studied only acidity changes seem to be really significant. Other parameters presented in Table 1 in most cases demonstrate low valuable development during germination. I should pay special attention to the ash data. Ash is known to reflect the content of minerals and I wonder how it is possible to be changed during germination without additional source of elements. That contradicts with the mass conservation law.
  2. As for protein content low variability of this parameter is of no doubt because to reflect intensive changes the authors ought to have analyzed water soluble protein
  3. The similar picture may be found for carbohydrate content where the greatest changes are to be expected for monosaccharides due to intensive hydrolyses.
  4. The authors should check statistics in Table 1 (and Table 2). For instance, the data for ash content in beans (3.6-3.4-3.0) the significance of differences according to the presented results should be 3.6a; 3.4b and 3.0c white the authors write: ‘3.6a; 3.4a and 3.0a’. Similar inconsistency are found in most of other cases. No statistics is presented for the acidity data where one should expect p<0.001 (???). By the way, for instance does ‘3.4±0.07’ indicate M±SD data?
  5. While reading the manuscript I can’t understand extremely high changes during germination in sodium content. Why sodium content increases during germination? Where does it come from??? Explanation is necessary. The same with Mg and to a lesser extent with Fe and Zn.
  6. The authors indicate that ‘Sodium, magnesium, iron and zinc availability increases for all the seeds samples  during the germination period’ which is in accordance of [48,49] data. Certainly minerals bioavailability increases during germination but total mineral content does not reflect the availability of this or that element. May be that should be water soluble forms that are precisely connected with bioavailability? But it seems that the presented data indicate only the total content. In this respect it is not right to include a statement in the conclusion; ‘Mineral contents (sodium, magnesium, zinc, iron) of the germinated legume seed increased during germination in all legume seed showing the beneficial influences of germination on mineral  bio accessibility’.
  7. Is it possible to achieve any correlations between the parameters tested )protein, fat, carbohydrates, minerals, etc)?

Minor comments

There are several misprints and inaccuracies:

  • line 62 : ‘Figure 1 shown the images’- change to ‘Figure 1 shows the images’
  • Lines 83,85.90 ‘In the four…’ change to ‘in the fourth’
  • Line 85 ‘it may clearly see’ change to ‘it may be clearly seen’
  • Lines 91-92 ‘The size of the radicle achieved on this day of germination the 24 mm length’- there is no predicate
  • Lines 102-104 ‘In the case of soybean and lupine seeds which contains less starch were visible more a dense compactness of the protein agglomerates’ revise the sentence (predicate (were visible) should be placed after the subject (dense compactness).
  • Line 105 ‘In germinated seeds may be seen significant structural changes in …’ change to ‘In germinated seeds significant structural changes may be seen in …’
  • Lines 124, 125, 220 ‘Shorter germination period leaded..’- check past tense
  • Line 134 ‘water from the germination process were removed’- change to ‘was removed’
  • Line 135 ‘clearly visible to all the germinated seed samples’; either ‘clearly visible to all of the germinated seed samples’, or ‘clearly visible in all the germinated seed samples’
  • Line 142 ‘are richer’- change to ‘is richer’
  • Line 146 ‘This may be due to the fact that during the germination process may have a hydrolysis of some proteins’ (change the words order and put the subject to the first place)
  • Line 160- the same: ‘for soybean were not noticed any significant variation’
  • Line 184 ‘it content did not significant changes’ change to ‘its content did not significant change’’
  • Line 198 ‘did not presented’ change to ‘did not present’
  • Line 309 ‘Previously studies indicates similar data’- use ‘indicate’
  • Line 328 ‘Previously of germination’ change to ‘Before the germination’
  • Line 364 ‘Physical-chemical’ change to ‘Physico-chemical’
  • Line 310 ‘CH3 and CH2’ change to ‘CH3 and CH2
  • Table 2 and in the text: ‘mg/100 g’ change to ‘mg 100 g-1

Author Response

9 March 2021

Dear Referee,  

We would like to thank the referee for the close reading and for the proper suggestions. We hope that we provide all the answers to the reviewer’s comments.

Thank you very much for the recommendations to publish our paper entitled “Impact of germination on the microstructural and physicochemical properties of different legumes types”.

The present version of the paper has been revised according to the reviewer’s suggestions.             

We uploaded the corrected version of the article for which we used the red color for the addition text.

GENERAL COMMENTS:

Referee comments: Among parameters studied only acidity changes seem to be really significant. Other parameters presented in Table 1 in most cases demonstrate low valuable development during germination. I should pay special attention to the ash data. Ash is known to reflect the content of minerals and I wonder how it is possible to be changed during germination without additional source of elements. That contradicts with the mass conservation law.

Response: We would like to thank to the referee for its close reading of our manuscript data. We were also surprised by the data obtained but we are sure that are the correct one. We have also did a flour analysis and on an automatic analyzer on which the ash content increases too.. We read a lot of article from the international literature and depending on the grain type, germination period and germination conditions the ash may vary by an increase or a decrease or may not present any significant changes. We find two main explanations about the possibility of ash increases (explanations that we incorporated in the manuscript):

  • The ash increase could be due to the increase in phytase activity during germination;
  • The ash increase may be due to the reduction of total soluble solids.

More we found studies (included in our manuscript) that reported that to 72 h of germination period the ash content was in the highest amount in the seed sample. After, the ash content begins to decrease – to see for example the article Chinma et al. (2009), Effect of germination on the chemical, functional and pasting properties of flour from brown and yellow varieties of tigernut (Cyperus esculentus) published in Food Research International, or  Echendu et al. (2009), Effects of Germination on Chemical Composition of Groundbean (Kerstingiella geocarpa harm) Seeds published in Pakistan Journal of Nutrition.

Some examples of studies (from Springer and Elsevier databases) which reported an ash increase during germination: Cornejo, F.; Novillo, G.; Villacrés, E.; Rosell, C. M. Evaluation of the physicochemical and nutritional changes in two amaranth species (Amaranthus quitensis and Amaranthus caudatus) after germination. Food Res. Int. 2019, 121, 933-939; Uppal and Bains (2012) Effect of germination periods and hydrothermal treatments on in vitro protein and starch digestibility of germinated legumes, J. Food Sci Technol-Mysore; 43;          Chinma, C.E.; Adedeji, O.E.; Etim, I.I.; Aniaka, G.I.; Mathew, E.O.; Ekeh, U.B.; Anumba, N.L.Physicochemical, nutritional, and sensory properties of chips produced from germinated African yam bean (Sphenostylisstenocarpa). LWT-Food Sci.Techol. 2021, 136, 110330, etc.; Huang. P.V.; Yen, N.T.H.; Phi, N.T.L.; Tien, N.P.H.; Trung, N.T.T. Nutritional composition, enzyme activities and bioactive compounds ofmung bean (Vignaradiata L.) germinated under dark and light conditions. LWT-Food Sci.Techol. 2020, 133, 110100

Referee comments: As for protein content low variability of this parameter is of no doubt because to reflect intensive changes the authors ought to have analyzed water soluble protein; The similar picture may be found for carbohydrate content where the greatest changes are to be expected for monosaccharides due to intensive hydrolyses.

Response: We would like to thank to the referee for its close reading of our manuscript data. We agree with his/her comments. We offer now in a more detailed way in the manuscript more discussion related to the reasons of protein and carbohydrate variability.

Referee comments: The authors should check statistics in Table 1 (and Table 2). For instance, the data for ash content in beans (3.6-3.4-3.0) the significance of differences according to the presented results should be 3.6a; 3.4b and 3.0c white the authors write: ‘3.6a; 3.4a and 3.0a’. Similar inconsistency are found in most of other cases. No statistics is presented for the acidity data where one should expect p<0.001 (???). By the way, for instance does ‘3.4±0.07’ indicate M±SD data?

Response: We would like to thank to the referee for its the close reading of our manuscript. We agree with the referee point of view. All the statistics part was revised. We hope now is oK. Yes, 3.4±0.07’ indicates  M±SD data. We missed the statistical part for acidity and pH values but we revised now. We really thank to the referee for noticed that.

Referee comments: While reading the manuscript I can’t understand extremely high changes during germination in sodium content. Why sodium content increases during germination? Where does it come from??? Explanation is necessary. The same with Mg and to a lesser extent with Fe and Zn.

Response: We did not found specifical explanations related to every mineral increase during the germination process only related to their increase to a generall level. We completed too, more informations in the manuscript related to mineral increase to a generall level.

Referee comments: The authors indicate that ‘Sodium, magnesium, iron and zinc availability increases for all the seeds samples  during the germination period’ which is in accordance of [48,49] data. Certainly minerals bioavailability increases during germination but total mineral content does not reflect the availability of this or that element. May be that should be water soluble forms that are precisely connected with bioavailability? But it seems that the presented data indicate only the total content. In this respect it is not right to include a statement in the conclusion; ‘Mineral contents (sodium, magnesium, zinc, iron) of the germinated legume seed increased during germination in all legume seed showing the beneficial influences of germination on mineral  bio accessibility’.

Response: We agree with the referee  point of view. We change now the phrase with another one as: Mineral contents (sodium, magnesium, zinc, iron) of the germinated legume seeds increased during germination in all legume seeds, showing the beneficial influences of germination on the nutritional profile of legumes.

Referee comments: Is it possible to achieve any correlations between the parameters tested ) protein, fat, carbohydrates, minerals, etc)?

Response: We made a PCA analysis to underline different correlations between the parameters tested.

MINOR COMMENTS:

There are several misprints and inaccuracies:

Referee comments: line 62 : ‘Figure 1 shown the images’- change to ‘Figure 1 shows the images’

Response: We changed.

Referee comments: Lines 83,85.90 ‘In the four…’ change to ‘in the fourth’

Response: We changed.

Referee comments: Line 85 ‘it may clearly see’ change to ‘it may be clearly seen’

Response: We changed.

Referee comments: Lines 91-92 ‘The size of the radicle achieved on this day of germination the 24 mm length’- there is no predicate

Response: We put now the predicat in the phrase.

Referee comments: Lines 102-104 ‘In the case of soybean and lupine seeds which contains less starch were visible more a dense compactness of the protein agglomerates’ revise the sentence (predicate (were visible) should be placed after the subject (dense compactness).

Response: We revised.

Referee comments: Line 105 ‘In germinated seeds may be seen significant structural changes in …’ change to ‘In germinated seeds significant structural changes may be seen in …’

Response: We changed.

Referee comments: Lines 124, 125, 220 ‘Shorter germination period leaded..’- check past tense

Response: We modified.

Referee comments: Line 134 ‘water from the germination process were removed’- change to ‘was removed’

Response: We changed.

Referee comments: Line 135 ‘clearly visible to all the germinated seed samples’; either ‘clearly visible to all of the germinated seed samples’, or ‘clearly visible in all the germinated seed samples’

Response: We changed.

Referee comments: Line 142 ‘are richer’- change to ‘is richer’

Response: We changed.

Referee comments: Line 146 ‘This may be due to the fact that during the germination process may have a hydrolysis of some proteins’ (change the words order and put the subject to the first place)

Response: We changed.

Referee comments: Line 160- the same: ‘for soybean were not noticed any significant variation’

Response: We changed.

Referee comments: Line 184 ‘it content did not significant changes’ change to ‘its content did not significant change’’

Response: We changed.

Referee comments: Line 198 ‘did not presented’ change to ‘did not present’

Response: We changed.

Referee comments: Line 309 ‘Previously studies indicates similar data’- use ‘indicate’

Response: We used indicate.

Referee comments: Line 328 ‘Previously of germination’ change to ‘Before the germination’

Response: We changed.

Referee comments: Line 364 ‘Physical-chemical’ change to ‘Physico-chemical’

Response: We changed.

Referee comments: Line 310 ‘CH3 and CH2’ change to ‘CH3 and CH2’

Response: We changed.

Referee comments: Table 2 and in the text: ‘mg/100 g’ change to ‘mg 100 g-1 ’

Response: We modified in the table.

Reviewer 2 Report

Below are my concerns about this work.

  1. Introduction lacks hypothesis and is poorly written.
  2. The statistical design should be factorial?
  3.  What is the posthoc analysis used that should be clearly mentioned? Moreover, the different varieties of each legume should be compared..
  4. Check for English language use and grammatical mistakes. 
  5. Elaborate the discussion section and discuss results thoroughly, not superficially.

Author Response

9 March 2021

Dear Referee,  

We would like to thank the referee for the close reading and for the proper suggestions. We hope that we provide all the answers to the reviewer’s comments.

Thank you very much for the recommendations to publish our paper entitled “Impact of germination on the microstructural and physicochemical properties of different legumes types”.

The present version of the paper has been revised according to the reviewer’s suggestions.             

We uploaded the corrected version of the article for which we used the red color for the addition text.

 Reviewer comments:

 Referee comments: Introduction lacks hypothesis and is poorly written. 

Response: In order to comply with the referee point of view we improved our introduction part. We wrote more informations related to our subject and we hope that now the introduction part to be more proper for the referee point of view.   Referee comments: The statistical design should be factorial?

Response: We agree with the referee point of view. We revised all the statistical part of our manuscript. 

Referee comments: What is the posthoc analysis used that should be clearly mentioned? Moreover, the different varieties of each legume should be compared.

Response: We agree with the referee point of view. Now we used the posthoc analysis and we compared the different varieties of each legume type. 

Referee comments: Check for English language use and grammatical mistakes. 

Response: The article was now revised by an English teacher.

 Referee comments: Elaborate the discussion section and discuss results thoroughly, not superficially.

Response: We separated now in the article the part of the results and discussion in results part (section 2) and discussion part (section 3).    
